# Individualized PEEP without Recruitment Maneuvers Improves Intraoperative Oxygenation: A Randomized Controlled Study

**DOI:** 10.3390/bioengineering10101172

**Published:** 2023-10-09

**Authors:** Lili Pan, Li Yang, Lingling Gao, Zhanqi Zhao, Jun Zhang

**Affiliations:** 1Department of Anesthesiology, Fudan University Shanghai Cancer Center, Department of Oncology, Shanghai Medical College, Fudan University, Shanghai 200032, China; pllsyf@163.com (L.P.); liyangmagic@sina.com (L.Y.); linglinggao@aliyun.com (L.G.); 2School of Biomedical Engineering, Guangzhou Medical University, Guangzhou 511436, China; 3Institute of Technical Medicine, Furtwangen University, 78054 Villingen-Schwenningen, Germany

**Keywords:** positive end-expiratory pressure, electrical impedance tomography, PaO_2_/FiO_2_, oxygenation, robot-assisted laparoscopic prostatectomy

## Abstract

Individualized positive end-expiratory pressure (PEEP) combined with recruitment maneuvers improves intraoperative oxygenation in individuals undergoing robot-assisted prostatectomy. However, whether electrical impedance tomography (EIT)-guided individualized PEEP without recruitment maneuvers can also improve intraoperative oxygenation is unknown. To test this, fifty-six male patients undergoing elective robot-assisted laparoscopic prostatectomy were randomly assigned to either individualized PEEP (Group PEEP_IND_, *n* = 28) or a control with a fixed PEEP of 5 cm H_2_O (Group PEEP_5_, *n* = 28). Individualized PEEP was guided by EIT after placing the patients in the Trendelenburg position and performing intraperitoneal insufflation. Patients in Group PEEP_IND_ maintained individualized PEEP without intermittent recruitment maneuvers, and those in Group PEEP_5_ maintained a PEEP of 5 cm H_2_O intraoperatively. Both groups were extubated in a semi-sitting position once the extubation criteria were met. The primary outcome was arterial oxygen partial pressure (PaO_2_)/inspiratory oxygen fraction (FiO_2_) prior to extubation. Other outcomes included intraoperative driving pressure, plateau pressure and dynamic, respiratory system compliance, and the incidence of postoperative hypoxemia in the post-operative care unit (PACU). Our results showed that the intraoperative median for PEEP_IND_ was 16 cm H_2_O (ranging from 12 to 18 cm H_2_O). EIT-guided PEEP_IND_ was associated with higher PaO_2_/FiO_2_ before extubation compared to PEEP_5_ (71.6 ± 10.7 vs. 56.8 ± 14.1 kPa, *p* = 0.003). Improved oxygenation extended into the PACU with a lower incidence of postoperative hypoxemia (3.8% vs. 26.9%, *p* = 0.021). Additionally, PEEP_IND_ was associated with lower driving pressures (12.0 ± 3.0 vs. 15.0 ± 4.4 cm H_2_O, *p* = 0.044) and better compliance (44.5 ± 12.8 vs. 33.6 ± 9.1 mL/cm H_2_O, *p* = 0.017). Our data indicated that individualized PEEP guided by EIT without intraoperative recruitment maneuvers also improved perioperative oxygenation in patients undergoing robot-assisted laparoscopic radical prostatectomy, which could benefit patients with the risk of intraoperative hemodynamic instability caused by recruitment maneuvers. Trial registration: China Clinical Trial Registration Center Identifier: ChiCTR2100053839. This study was registered on 1 December 2021. The first patient was recruited on 15 December 2021.

## 1. Introduction

Robot-assisted laparoscopic radical prostatectomy is associated with high abdominal pressure and requires an extreme Trendelenburg position for optimal surgical access [1]. During general anesthesia, these two factors combined with lung ventilation shifting toward ventral regions promote pulmonary atelectasis, which can contribute to postoperative pulmonary complications [2]. Because of a patient’s individual constitution, body mass index (BMI), positioning, and intra-abdominal pressure, the individual positive end-expiratory pressure (PEEP) setting during intraoperative mechanical ventilation is recommended to counteract these effects [3,4,5]. The individual PEEP setting during general anesthesia has been reported to reduce intrapulmonary shunt and postoperative pulmonary complications in elderly patients undergoing spinal surgery in the prone position [6].

Several techniques have been used to determine individual PEEP, also known as optimal PEEP [7,8,9]. Chest computer tomography (CT) is the gold standard technique for the assessment of lung inflation and atelectasis [10]. However, it exposes individuals to radiation and is not feasible for bedside use. The measurement of transpulmonary pressure is another alternative that can be used at the bedside, although it requires special training and additional equipment in order to obtain transpulmonary pressure [11]. However, this method is not practical for evaluating regional lung ventilation distribution. Electrical impedance tomography (EIT) has been demonstrated to individualize the PEEP level [12,13], which minimizes atelectasis and prevents lung overdistension [14,15]. Individual PEEP guided by EIT and combined with recruitment maneuvers improves intraoperative oxygenation, compared with standard ventilation with PEEP of 5 cm H_2_O, both in morbidly obese and normal patients undergoing laparoscopic surgery [4,16]. However, recruitment maneuvers may cause intraoperative hemodynamic instability [4,17] and intermittent manual operations are inconvenient.

Therefore, we tested the hypothesis that EIT-guided individualized PEEP without recruitment maneuvers also improves intraoperative oxygenation, when compared to 5 cm H_2_O PEEP in patients undergoing robot-assisted laparoscopic radical prostatectomy.

## 2. Materials and Methods

### 2.1. Ethics Approval

This prospective, randomized, controlled, single-center study was approved by the ethics committee of Fudan University Shanghai Cancer Center (number: IRB2010225-11) and registered with the China Clinical Trial Registration Center (ChiCTR2100053839). The first patient was recruited on 15/12/2021 by the Fudan University Shanghai Cancer Center. Informed consent was obtained from every patient before enrollment. An investigator assessed patients for eligibility the day before surgery.

### 2.2. Inclusion Criteria

Patients undergoing elective robot-assisted prostate surgery under general anesthesia;American Society of Anesthesiologists (ASA) grade I~III;BMI < 30 kg/m^2^.

### 2.3. Exclusion Criteria

Untreated ischemic heart disease;Acute or chronic respiratory failure and moderate to severe obstructive or restrictive pulmonary diseases, including chronic obstructive pulmonary disease and asthma;Previous lung or airway surgery;Neuromuscular diseases;Preoperative SpO_2_ (in room air) < 95%.

### 2.4. Anesthesia Management

The demographic data and clinical characteristics were recorded after patients entered the operating room. Standard monitoring protocols were applied, including ECG, pulse oximetry, and capnography. Intravenous access was established, and a radial arterial line was established for invasive blood pressure monitoring and arterial blood gas (ABG) analysis. Pre-oxygenation was performed for 3 min (oxygen concentration: 100%, oxygen flow: 6 L/min) before anesthesia. The anesthetic induction was conducted with an intravenous-targeted control infusion (TCI) of propofol (Marsh mode) at 4 μg/mL, sufentanil at 0.3 μg/kg, remifentanil (Minto mode) at 2 ng/mL, and rocuronium at 0.6 mg/kg. A 7.0-size tracheal tube was inserted, and correct placement was confirmed through auscultation and the presence of bilateral equal breath sounds. General anesthesia was maintained with the continuous TCI infusion of propofol 3–4 μg/mL and remifentanil 1–2 ng/mL, as well as the intermittent administration of rocuronium in order to maintain adequate muscle paralysis.

### 2.5. Intraoperative Ventilation

After tracheal intubation, patients were provided with pressure-regulated volume-controlled ventilation (Flow-I, Maquet Inc., Heidelberg, Germany). The ventilation was set at a tidal volume of 6–8 mL/kg predicted body weight (PBW), FiO_2_ = 0.5, PEEP = 18 cm H_2_O in the PEEP_IND_ Group (before individual PEEP titration), and a respiratory rate of 12–15 beats/min was set to maintain the end-tidal CO_2_ partial pressure between 35 and 45 mmHg.

### 2.6. Measurements

Demographic data and medical characteristics, including sex, age, BMI, PBW, ASA classification, medical history, invasive blood pressure, and heart rate, were collected. Intraoperative tidal volume, dynamic respiratory system compliance (Cdyn), driving pressure, plateau pressure, and ABG at 5 and 60 min after an individual or fixed PEEP setting and before extubation were recorded. The ABG analysis (GEM3500; Instrumentation Laboratory, USA) was performed simultaneously. We collected regional lung ventilation distribution before anesthetic induction (T_0_) 5 and 60 min after the PEEP setting (T_1_, T_2_) and 10 min after extubation (T_3_).

EIT data were obtained using a commercial EIT system (PulmoVista500; Draeger Medical, Luebeck, Germany) after PEEP levels had been set in both groups. In the present study, an EIT electrode belt, which carries 16 electrodes with a width of 40 mm, was placed around the thorax in the fifth intercostal space, and a reference electrode was placed on the right thorax.

### 2.7. The Individual PEEP Titration by EIT

All patients were placed in a steep 30-degrees Trendelenburg position prior to incision. Abdominal access, insufflation, and docking of the robot occurred at a pneumoperitoneum pressure of 14 mmHg. All cases were completed using the Davinci^TM^ Xi Robot and AirSeal^TM^ iFS device (Intuitive Surgical, Inc., Sunnyvale, CA, USA). The individual PEEP titration with EIT in the PEEP_IND_ Group started after the patient’s placement in the Trendelenburg position and peritoneal insufflation (Appendix A). PEEP started at 18 cm H_2_O and decreased by 2 cm H_2_O each time for 2 min until 8 cm H_2_O was reached. Then, the optimal PEEP value was defined as the intercept of cumulated collapse and overdistension percentage curves to minimize regional compliance loss [18] using customized software [19]. After individual PEEP titration, patients in the PEEP_IND_ Group were maintained at their individual PEEP level until extubation; by contrast, those in the PEEP_5_ Group were maintained at a fixed PEEP level (5 cm H_2_O) until extubation. There were no intermittent recruitment maneuvers in either group.

### 2.8. Outcome Measures

The primary outcome in the present study was PaO_2_/FiO_2_ before extubation. Other outcome parameters included:(1)Intraoperative driving pressure, plateau pressure, and Cdyn;(2)Ventral over dorsal ventilation distribution ratio;(3)Intraoperative hemodynamics;(4)The incidence of postoperative hypoxemia (SpO_2_ < 92% were on room air after extubation) in the PACU.

### 2.9. Statistical Analysis

Intraoperative PaO_2_/FiO_2_ was reported as 55.7 ± 10.8 kPa before extubation in patients undergoing robot-assisted laparoscopic prostatectomy [4]. We assumed that there was a 9 kPa difference between the two groups, with a variance of 10.8 kPa, a statistical power of 80%, and a two-sided α significance level of 0.05; therefore, at least 23 patients in each group did not need to be recruited. Considering a dropout rate of 20%, a total of 29 patients for each group (for a total of 58 patients) were enrolled. A minimization randomization method was performed via MinimPy2 software (version 2.0, OSDN, Columbus, OH, USA). SPSS 20.0 software was used for statistical analysis. Measurement data were expressed as mean ± standard deviation or median (interquartile range) where appropriate. The Chi-square test and Fisher’s exact test were used for categorical variables, and the *t*-test or Mann–Whitney U test were used for continuous variables. The differences were considered statistically significant when *p* < 0.05.

## 3. Results

A total of 58 patients were initially enrolled in the study period. Fifty-six patients were randomized (Figure 1). Finally, 52 patients were included in the final analysis, 26 in each group. Their clinical characteristics are listed in Table 1.

The mean individualized PEEP guided by EIT in the PEEP_IND_ Group was 16 cm H_2_O (ranging from 12 to 18 cm H_2_O).

PaO_2_/FiO_2_ in the PEEP_IND_ Group was 14.8 kPa higher than the PEEP_5_ Group before extubation (71.6 ± 10.7 vs. 56.8 ± 14.1 kPa, *p* = 0.003, Table 2).

Before extubation, the individualized PEEP level guided by EIT was associated with higher plateau pressure when compared to the 5 cm H_2_O PEEP setting (27.5 ± 3.4 cm H_2_O vs. 20.0 ± 4.4 cm H_2_O, *p* < 0.001), resulting in significantly lower driving pressures (12.0 ± 3.0 cm H_2_O vs. 15.0 ± 4.4 cm H_2_O, *p* = 0.044) and better Cdyn (44.5 ± 12.8 cm H_2_O vs. 33.6 ± 9.1 cm H_2_O, *p* = 0.017). The results are shown in Table 2. The power test of the driving pressure was 0.94, plateau pressure was 1, and Cdyn was 0.99.

The hemodynamic parameters, such as heart rate, blood pressure, doses of vasopressor, and fluid infusions, during mechanical ventilation were similar between the two groups (Table 1 and Table 2).

Further analysis indicated that individualized PEEP guided by EIT was associated with improved regional lung ventilation distribution when compared with 5 cm H_2_O PEEP during mechanical ventilation (Table 3), which resulted in improved oxygenation immediately after extubation and a lower incidence of postoperative hypoxemia in the PACU (3.8% vs. 26.9%, *p* = 0.021, Table 4).

## 4. Discussion

In the present study, we found that maintaining the individualized PEEP level guided by EIT without recruitment maneuvers improved intraoperative oxygenation. The benefit of intraoperative individualized PEEP could extend into the postoperative period after extubation, as evidenced by an increase in PaO_2_/FiO_2_ and a decrease in the incidence of postoperative hypoxemia in PACU when compared with a fixed PEEP setting.

We tested our hypothesis on patients who underwent robot-assisted laparoscopic radical prostatectomy. Robot-assisted radical prostatectomy can be performed via either transperitoneal or total extra-peritoneal approaches. Transperitoneal robot-assisted radical prostatectomy, the most frequently used approach worldwide, has the advantages of a wider working space, a shorter duration for trocar positioning and preparing the workspace, and meticulous lymph node dissection [20,21,22]. The disadvantages of transperitoneal robot-assisted radical prostatectomy are associated with a higher risk of bowel injury, contraindications for obesity or intraperitoneal adhesions, and Trendelenburg positioning at nearly 35–40 degrees [20,21,22]. However, with no contact with the bowel and less inclined patient positioning (15–20 degrees), extra-peritoneal robot-assisted radical prostatectomy has the technical disadvantages of a narrower working space, more time spent on preparing the extraperitoneal field, and the lack of a combined approach to the bladder anteriorly and posteriorly [20,21]. In addition, extra-peritoneal robot-assisted radical prostatectomy is associated with higher partial CO_2_ pressure, causing a subsequent decrease in arterial pH [21]. In our medical center, robot-assisted radical prostatectomy is commonly performed via transperitoneal approaches.

High degrees of Trendelenburg positioning may cause peripheral nerve injuries [23,24] and increase the airway pressure, causing pneumothorax. Fortunately, no pneumothorax occurred in our study. The reasons for this may be that relatively low degrees of the Trendelenburg position (approximately 30 degrees in our study) were used, and patients with previous respiratory diseases were excluded. In addition, different degrees of the Trendelenburg position might lead to different individual PEEPs measured using EIT. Therefore, we used the same degree of the Trendelenburg position in the two study groups to avoid any confounding effects on body position.

The use of pneumoperitoneum is needed during robot-assisted radical prostatectomies, although it is a known risk given the change in physiological parameters that accompany its utilization. Conventionally, 15 mmHg is used in robot-assisted radical prostatectomy. However, it has also been reported that decreasing insufflation pressures from 15 to 12 mmHg can decrease the length of stay and postoperative ileus rates [25]. We used 14 mmHg pneumoperitoneum in our study.

The Trendelenburg position and high abdominal pressure may impede mechanical ventilation, leading to perioperative atelectasis in laparoscopic surgeries. Previous studies have demonstrated that using individualized PEEP combined with intermittent recruitment maneuvers for laparoscopic surgeries can counteract atelectasis formation and improve oxygenation [14]. The individualized PEEP identified by EIT has the substantial advantage of being used at the bedside [26,27]. The individualized PEEP titration process could be influenced by surgical maneuvers; therefore, in order to avoid this influence in the present study, surgeons were asked to suspend their surgical procedures during PEEP titration. The duration of PEEP titration was about 30 min. Since the surgical robot needed some time to prepare after patients had been placed in the Trendelenburg position, the actual time that it cost surgeons waiting to perform the operation was limited (5–10 min at most).

Intermittent recruitment maneuvers were not combined with individualized PEEP in our study. Girrbach et al. applied recruitment maneuvers and individualized PEEP to significantly improve intraoperative oxygenation [4]. However, hemodynamic instability is one of the concerns during recruitment maneuvers [17]. Since most cardiovascular complications, such as bradycardia and hypotension, occur during recruitment maneuvers, the majority of patients require vasopressor support to maintain normal blood pressure. Without this intermittent recruitment maneuver, medics may worry that using the same PEEP setting during the operation could lead to lung collapse. Interestingly, our results demonstrate that individualized PEEP without recruitment maneuvers also improves intraoperative oxygenation. Futier et al. compared PEEP alone and PEEP combined with recruitment maneuvers in normal weight and obese patients, respectively, undergoing laparoscopic surgery, and the results showed that 10 cm of H_2_O PEEP without recruitment maneuvers did not improve oxygenation, while the addition of a recruitment maneuver increased lung elastane and oxygenation [28]. Further, intraoperative-fixed PEEP level may lead to either lung overinflation or atelectasis [29]. The individualized PEEP level could improve respiratory outcomes better than fixed PEEP without increasing lung complications [4,30,31,32]. This could explain the discrepancy between the findings of Futier et al. and our own results. Shono et al. showed that high PEEP (15 cm H_2_O) can be associated with higher lung ventilation and oxygenation in prostate patients undergoing robot-assisted endoscopic procedures with their heads below 25° [33]. At the end of the operation, a moderate PEEP level without lung recruitment maneuvers was sufficient to achieve substantial lung opening. Our study’s findings may benefit patients with a risk of intraoperative hemodynamic instability. It also indicates that recruitment maneuvers may be unnecessary to improve oxygenation.

Our data show that a high incidence of postoperative hypoxemia (26.9% in the control group) was found after robot-assisted prostatectomy, which extended the PACU stay and increased costs. With the implementation of EIT-guided PEEP titration, PaO_2_/FiO_2_ was 14.8 kPa higher than the control group before extubation (71.6 ± 10.7 vs. 56.8 ± 14.1 kPa, *p* = 0.003, Table 2). The improved oxygenation extended into the PACU, as evidenced by a lower incidence of postoperative hypoxemia (3.8% vs. 26.9%, *p* = 0.021). Additionally, EIT-guided PEEP titration was associated with lower driving pressures and better compliance. These improvements could be explained by the distribution of local lung ventilation. These results indicate that postoperative pulmonary complications can be improved in such patients, and prognosis might also be improved. Though we did not follow up with postoperative pulmonary complications, the feasibility and advantages of using EIT in individualized PEEP titration undergoing robot-assisted prostatectomy were demonstrated.

High-level PEEP may aggravate hemodynamic instability, especially in critically ill patients [34,35]. However, there were no significant differences between the vasopressor doses used in both groups in our study, indicating that individualized PEEP guided by EIT might not significantly influence hemodynamics, which is consistent with a previous study [31]. Our results demonstrate that the benefits of intraoperative individual PEEP are sustained immediately after surgery, which is consistent with a previous study in which intraoperative individualized PEEP reduces postoperative atelectasis [36]. However, whether intraoperative PEEP improves postoperative oxygenation is controversial. Several studies have also reported that intraoperative PEEP only improved intraoperative oxygenation and failed to maintain its effect into the postoperative period [4,16]. This discrepancy may be due to different extubation approaches in different surgery centers. In our hospital, it is a routine practice to perform tracheal extubation in a semi-sitting position, which is associated with better functional residual capacity in comparison to the supine position. However, this notion requires further validation.

Our results suggest that improved regional lung ventilation distribution, as evidenced by a better ventral/dorsal distribution ratio, might contribute to improved perioperative oxygenation using EIT-guided individualized PEEP. The ventral/dorsal distribution ratio was calculated based on the percentage of tidal variation in regions of interest, and it was easily calculated by bedside personnel [37]. Patients under mechanical ventilation are usually associated with shunts due to alveolar collapse in the dorsal lung (as shown by the ventral/dorsal ratio increases, Table 3). Alveolar collapse is usually accompanied by an intrapulmonary shunt, resulting in impaired pulmonary gas exchange [38]. The significant effect of lung volume reduction on pulmonary dynamics promotes the development of atelectasis [39].

The application of EIT technology requires specialized training, and the equipment is expensive, which may require the consideration of its cost-effectiveness. However, our results show that EIT technology can improve intraoperative oxygen and decrease the incidence of postoperative hypoxemia, which may shorten patients’ hospital stays and decrease postoperative comorbidity. The EIT examination itself generates almost no cost (i.e., no consumables and low maintenance cost). The overall cost-effectiveness could be in favor of applying EIT-guided PEEP, especially for high-risk patients prone to lung atelectasis.

There are several limitations to this study. Firstly, individualized PEEP guided by EIT limited the regions of overdistension and collapse, which might not be the “optimal” PEEP in other aspects, e.g., transpulmonary pressure. The individualized PEEP titration ranged from 18 cm H_2_O to 8 cm H_2_O. If optimal PEEP was outside this range, which would be rare, our method would not be able to find it. Nevertheless, the PEEP values achieved in our study resulted in better perioperative oxygenation. However, since the sample size was small, the effect of individualized PEEP on postoperative pulmonary complications, such as atelectasis and pneumonia, was not determined in the present study.

## 5. Conclusions

Compared with a fixed PEEP level, the individualized PEEP level achieved using EIT without recruitment maneuvers could also improve perioperative oxygenation in patients undergoing robot-assisted laparoscopic radical prostatectomy, which may benefit patients at risk of intraoperative hemodynamic instability caused by recruitment maneuvers. However, whether this individualized PEEP without intermittent recruitment maneuvers can reduce the incidence of postoperative pulmonary complications should be further studied.

## Figures and Tables

**Figure 1 bioengineering-10-01172-f001:**
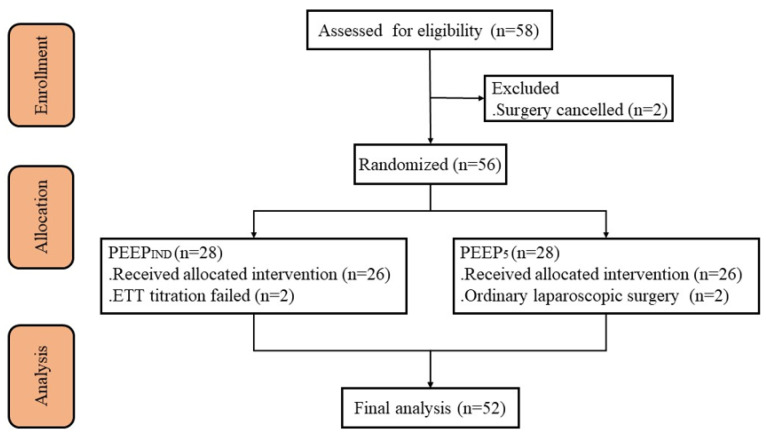
Flowchart of patient enrollment. PEEP_IND_, individualized PEEP guided by electrical impedance tomography; PEEP_5_, standard PEEP setting of 5 cm H_2_O.

**Table 1 bioengineering-10-01172-t001:** Baseline characteristics of the study population.

Characteristic	PEEP_5_ (*n* = 26)	PEEP_IND_ (*n* = 26)
Age (yr)	65 (52–83)	67.8 (53–80)
<65 [*n* (%)]	13 (50)	9 (34.6)
BMI (kg m^−2^)	24.4 (2.1)	24.1 (2.2)
ASA physical status		
I	3 (11.5)	4 (15.4)
II	22 (84.6)	22 (84.6)
III	1 (3.8)	0 (0)
Hypertension (%)	12 (46.2)	7 (26.9)
Diabetes (%)	3 (11.5)	3 (11.5)
Total fluid infusion (mL)	2063.5 ± 257.1	1932.7 ± 342
Blood loss (mL)	124.2 ± 50.5	123.1 ± 45.2
Urinary output (mL)	357.7 ± 177	330.8 ± 228.1
Vasoactive injections		
Ephedrine (mg)	6.7 ± 4.7	7.4 ± 6.6
Phenylephrine (μg)	204.6 ± 223.7	127.7 ± 176.5
Anesthesia duration (min)	227.4 ± 35.1	218 ± 34.1
Surgery duration (min)	169.4 ± 29.3	169.6 ± 27.4
PEEP (cm H_2_O)	5 (5–5)	16 (12–18)

Data are presented as mean ± SD, mean (range), or number (%). ASA, American Society of Anesthesiologists; BMI, body mass index; PEEP, positive end-expiratory pressure.

**Table 2 bioengineering-10-01172-t002:** Respiratory and hemodynamic parameters during mechanical ventilation.

Parameter	Time Points	PEEP_5_ (*n* = 26)	PEEP_IND_ (*n* = 26)	*p*-Value
PaO_2_/FiO_2_ (kPa)	PEEP 5 min	56.6 ± 9.0	63.8 ± 14.2	0.033
	PEEP 60 min	57.3 ± 14.4	65.5 ± 13.1	0.036
	Pre-extubation	56.8 ± 14.1	71.6 ± 10.7	0.003
Plateau pressure (cm H_2_O)	PEEP 5 min	21.3 ± 5.0	27.9 ± 4.33	0.000
	PEEP 60 min	21.3 ± 3.0	27.6 ± 4.2	0.000
	Pre-extubation	20.0 ± 4.4	27.5 ± 3.4	0.000
Driving pressure (cm H_2_O)	PEEP 5 min	14.2 ± 3.0	13.7 ± 3.6	0.586
	PEEP 60 min	15.8 ± 2.6	13.4 ± 4.1	0.014
	Pre-extubation	15.0 ± 4.4	12.0 ± 3.0	0.044
Cdyn (mL/cm H_2_O)	PEEP 5 min	32.6 ± 6.1	35.9 ± 7.0	0.076
	PEEP 60 min	29.6 ± 5.2	37.5 ± 9.2	0.000
	Pre-extubation	33.6 ± 9.1	44.5 ± 12.8	0.017
HR	PEEP 5 min	63.0 ± 8.9	64.4 ± 8.7	0.562
	PEEP 60 min	66.2 ± 10.0	61.4 ± 8.9	0.076
	Pre-extubation	66.2 ± 8.1	60.2 ± 8.1	0.080
SBP	PEEP 5 min	110.6 ± 13.3	113.8 ± 12.0	0.363
	PEEP 60 min	114.9 ± 23.7	117.5 ± 12.6	0.632
	Pre-extubation	117.9 ± 12.3	123.8 ± 17.8	0.321
DBP	PEEP 5 min	59.8 ± 8.9	61.0 ± 6.9	0.579
	PEEP 60 min	60.3 ± 9.6	65.0 ± 10.2	0.088
	Pre-extubation	62.5 ± 8.3	65.9 ± 9.1	0.315

Data are presented as mean ± SD. PEEP 5 min, five minutes after PEEP setting; PEEP 60 min, 60 min after PEEP setting; Pre-extubation, before extubation but after operation; HR, heart rate; SBP, systolic blood pressure; DBP, diastolic blood pressure.

**Table 3 bioengineering-10-01172-t003:** Regional lung ventilation distribution.

	PEEP		Pw	Pb
	PEEP_5_ (*n* = 26)	PEEP_IND_ (*n* = 26)		
Ventral/dorsal distribution ratio				
T_0_	1.09 (0.97–1.21)	1.08 (0.96–1.19)	0.004	0.000
T_1_	1.39 (1.26–1.52)	0.99 (0.86–1.12)		
T_2_	2.00 (1.53–2.47)	1.02 (0.57–1.47)		
T_3_	1.27 (1.06–1.47)	1.17 (0.97–1.37)		

T_0_: before induction of anesthesia; T_1_: 5 min after PEEP setting; T_2_: 60 min after PEEP setting; T_3_: 10 min after extubation. Data are presented as means and 95% confidence interval two-factor repeated measures through ANOVA. Pw—within group interactions, Pb—between groups interactions with post hoc Bonferroni correction.

**Table 4 bioengineering-10-01172-t004:** Incidence of hypoxemia in the PACU.

	Total (*n*)	Hypoxemia [*n* (%)]	Non-Hypoxemia [*n* (%)]	*p*-Value
PEEP_5_	26	7 (26.9)	19 (73.1)	0.021
PEEP_IND_	26	1 (3.8)	25 (96.2)	

Data are presented as numbers (%). PEEP_IND_, individualized PEEP setting guided by electrical impedance tomography; PEEP_5_, standard PEEP setting of 5 cm H_2_O. PACU, post-anesthesia care unit.

## Data Availability

The datasets used and/or analyzed during the current study are available from the corresponding author on reasonable request.

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
