# Peer review of "Individualized PEEP without Recruitment Maneuvers Improves Intraoperative Oxygenation: A Randomized Controlled Study"

_bioengineering, 2023, doi:10.3390/bioengineering10101172_

Round 1
Reviewer 1 Report
The authors performed an excellent work on positive end-expiratory pressure (PEEP). They compared individualized PEEP (Group PEEPIND) with fixed PEEP of 5 cmH2O (Group PEEP5). Individualized PEEP was guided by electrical impedance tomography (EIT) after placing the patients in the Trendelenburg position and intraperitoneal insufflation. Individualized PEEP guided by EIT without intraoperative recruitment maneuvers improved perioperative oxygenation in patients undergoing robot-assisted radical prostatectomy.
The manuscript is very good but several improvements are needed:
- Please report the long form before the contracted version (ex line 51 chest CT, please write chest computer tomography)
- Are patients with previous lung or airway surgery included? Please specify
- Please develop the concept of the intraperitoneal technique that has been used for radical prostatectomy. This technique could have an impact on the results because CO2 reabsorption is lower with respect to the extraperitoneal but the Trendelenburg is higher. Please develop these themes (doi: 10.1590/S1677-5538.IBJU.2014.0199 and doi: 10.3390/app11041513)
- How many degrees of Trendelenburg? It could have an impact on the results as it could augment the risk of pneumothorax (doi: 10.3389/fsurg.2023.1157528)
- Which CO2 pressure for insufflation has been used? Please specify because it could have an impact on the results (doi: 10.1007/s00345-020-03486-4)
Author Response
Revision Notes
Dear editors and reviewers:
We would like to thank for your kind letter and for reviewers’ constructive comments on our manuscript entitled " Individualized PEEP without recruitment maneuvers improves intraoperative oxygenation: A randomized controlled study". These comments are very valuable and helpful for us to improve our manuscript. All authors have seriously discussed according to all these comments, thereby we have tried our best to revise our manuscript to meet the requirements of your journal. In this revised version, changes in our manuscript within the document were all highlighted in blue.
Please find our point-by-point responses to the comments as follows:
Referee: 1
Comment: Please report the long form before the contracted version (ex line 51 chest CT, please write chest computer tomography).
Response: Thanks for this nice suggestion. We have changed those in our manuscript according to your suggestions.
Line 54 “Chest computer tomography (CT)”
Line 83 “American Society of Anesthesiologists (ASA)”
Comment: Are patients with previous lung or airway surgery included? Please specify
Response: Thanks for this valuable comment. There were no patients with previous lung or airway surgery included in our study. We have added this as one of the exclusion criteria in our manuscript.
Line 94, exclusion criteria 3 “Previous lung or airway surgery”.
Comment: Please develop the concept of the intraperitoneal technique that has been used for radical prostatectomy. This technique could have an impact on the results because CO2 reabsorption is lower with respect to the extraperitoneal but the Trendelenburg is higher. Please develop these themes (doi: 10.1590/S1677-5538.IBJU.2014.0199 and doi: 10.3390/app11041513)
Response: Thanks for this nice suggestion. We have developed these themes in the discussion of our manuscript.
“We tested our hypothesis in patients who underwent robot-assisted laparoscopic radical prostatectomy. Robot-assisted radical prostatectomy can be performed via either transperitoneal or total extra-peritoneal approaches. Transperitoneal robot-assisted radical prostatectomy, the most frequently used approach worldwide, has advantages of wider working space, shorter duration for trocar positioning and preparation of working space, and meticulous lymph node dissection [20-22]. The disadvantages of transperitoneal robot-assisted radical prostatectomy are associated with higher risk of bowel injury, contraindications for obesity or intraperitoneal adhesions, and Trendelenburg positioning in nearly 35-40 degrees [20-22]. However, with no contact with the bowel and a less inclined patient positioning (15-20 degrees), extra-peritoneal robot-assisted radical prostatectomy has technical disadvantages of a narrower working space, spending more time for preparation of extraperitoneal field, and lack of a combined approach to the bladder anteriorly and posteriorly [20,21]. Besides, extra-peritoneal robot-assisted radical prostatectomy is associated with higher partial CO2 pressure causing a subsequent decrease in arterial pH [21]. In our center, robot-assisted radical prostatectomy is commonly performed via transperitoneal approaches.”
Akand M, Erdogru T, Avci E and Ates M 2015 Transperitoneal versus extraperitoneal robot-assisted laparoscopic radical prostatectomy: A prospective single surgeon randomized comparative study INT J UROL 22 916-21
Cochetti G, Del Zingaro M, Ciarletti S, Paladini A, Felici G, Stivalini D, Cellini V and Mearini E 2021 New Evolution of Robotic Radical Prostatectomy: A Single Center Experience with PERUSIA Technique Applied sciences 11 1513
Dal Moro F, Crestani A, Valotto C, Guttilla A, Soncin R, Mangano A and Zattoni F 2015 Anesthesiologic effects of transperitoneal versus extraperitoneal approach during robot-assisted radical prostatectomy: results of a prospective randomized study International Brazilian Journal of Urology 41 466-72
Comment: How many degrees of Trendelenburg? It could have an impact on the results as it could augment the risk of pneumothorax (doi: 10.3389/fsurg.2023.1157528)
Response: Thanks for this valuable suggestion. Approximately 30 degrees of Trendelenburg position was used in our study. We have added this information in our manuscript.
Line 132 “All patients were placed in steep 30 degrees Trendelenburg prior to incision.”
Furthermore, we discussed the influence of Trendelenburg degree on the results:
Line 230 “High degrees of Trendelenburg position may cause peripheral nerve injuries [23,24] and increase the airway pressure causing pneumothorax. Fortunately, no pneumothorax occurred in our study. The reasons may be that relatively low degrees of Trendelenburg position (approximately 30 degrees in our study) was used and patients with previous respiratory diseases were excluded. Besides, different degrees of Trendelenburg position might lead to different individual PEEP measured by EIT. Therefore, we used the same degrees of Trendelenburg position in two study groups, to avoid the confound of body position.”
Di Pierro G B, Wirth J G, Ferrari M, Danuser H and Mattei A 2014 Impact of a Single-surgeon Learning Curve on Complications, Positioning Injuries, and Renal Function in Patients Undergoing Robot-assisted Radical Prostatectomy and Extended Pelvic Lymph Node Dissection Urology (Ridgewood, N.J.) 84 1106-11
Paladini A, Cochetti G, Felici G, Russo M, Saqer E, Cari L, Bordini S and Mearini E 2023 Complications of extraperitoneal robot-assisted radical prostatectomy in high-risk prostate cancer: A single high-volume center experience Frontiers in surgery 10 1157528
Comment: Which CO2 pressure for insufflation has been used? Please specify because it could have an impact on the results (doi: 10.1007/s00345-020-03486-4)
Response: Thanks for your suggestion. CO2 pressure of 14 mmHg was used for insufflation in our study. We have specified this in our manuscript.
Line 133 “Abdominal access, insufflation and docking of the robot occurred at a pneumoperitoneum pressure of 14 mmHg.”
Line 239 “The use of pneumoperitoneum is needed during robotic assisted radical prostatectomies, which is a known risks given the change in physiological parameters that accompany its utilization. Conventionally, 15 mmHg is used in robotic assisted radical prostatectomy. However, it was also reported that decreasing insufflation pressures from 15 to 12 mmHg could decrease length of stay and postoperative ileus rates [25]. We used 14 mmHg pneumoperitoneum in our study.”
Rohloff M, Cicic A, Christensen C, Maatman T K, Lindberg J and Maatman T J 2019 Reduction in postoperative ileus rates utilizing lower pressure pneumoperitoneum in robotic-assisted radical prostatectomy Journal of robotic surgery 13 671-4
Best regards!
Reviewer 2 Report
I would like to thank the handling editor for offering me the opportunity to review the manuscript entitled “Individualized PEEP without recruitment maneuvers improves intraoperative oxygenation: A randomized controlled study” authored by Pan and colleagues, which is currently under consideration for publication in Bioengineering. I would also like to commend the authors for their scholarly work, which examines an important clinical topic.
In particular, the manuscript reports the results of a randomized controlled trial investigating the effects of individualized positive end-expiratory pressure (PEEP) guided by electrical impedance tomography (EIT) without recruitment manoeuvres on oxygenation in patients undergoing robot-assisted laparoscopic radical prostatectomy. Fifty-six patients were randomized to receive either individualized PEEP titrated by EIT (PEEPIND group) or standard PEEP of 5 cmH2O (PEEP5 group). The primary outcome was the arterial oxygen partial pressure (PaO2) to inspiratory oxygen fraction (FiO2) ratio prior to extubation. Secondary outcomes included intraoperative respiratory mechanics, hemodynamic parameters, and postoperative hypoxemia incidence. The results showed that patients in the PEEPIND group had significantly higher PaO2/FiO2 ratios compared to the PEEP5 group before extubation. The PEEPIND group also had lower driving pressures, better respiratory system compliance, more homogeneous lung ventilation distribution, and lower postoperative hypoxemia incidence. There were no differences in hemodynamic parameters between groups. The authors conclude that EIT-guided individualized PEEP without recruitment manoeuvres improves oxygenation during and after robot-assisted laparoscopic radical prostatectomy compared to standard low PEEP. The study provides evidence supporting the use of individualized PEEP titration to optimize intraoperative ventilation in this surgical population.
This randomized controlled trial appears to be scientifically, technically, and ethically sound based on the information provided in the manuscript. The study design and methodology seem appropriate to evaluate the research question on the effects of individualized PEEP on oxygenation. Randomization has been implemented to minimize bias, while the employed statistical analyses are valid. There are no overt ethical concerns noted.
The study's main strength is its randomized controlled design comparing an intervention of individualized PEEP titration by EIT to standard low PEEP in patients undergoing robotic surgery. This allows for direct comparison of the effects of the PEEP strategies on clinically relevant outcomes. Other strengths include the variety of respiratory outcomes examined and the inclusion of postoperative assessments.
The manuscript has the potential to make a meaningful contribution to the existing literature on individualized PEEP and intraoperative ventilation. While previous studies have evaluated EIT-guided PEEP with recruitment manoeuvres, this study specifically examines individualized PEEP alone. The results provide evidence that EIT-based PEEP titration can improve oxygenation even without recruitment manoeuvres. The originality of the study is the focus on patients undergoing robotic prostatectomy, a population in which optimal ventilation strategies are still being defined. If the results are confirmed, this could influence how PEEP is implemented during robotic surgery.
Overall, this rigorous clinical trial adds value by demonstrating the feasibility of individualized PEEP to improve oxygenation intraoperatively and postoperatively in this surgical setting. The results merit consideration for potential impact on improving ventilation management and reducing complications in patients undergoing robotic surgery. Additional studies to confirm generalizability would be beneficial.
While the manuscript provides valuable insights, there are some areas that could be refined to further augment the quality and impact of the work. Here are some respectful suggestions that could potentially improve the paper if the authors choose to implement them:
Abstract:
- The abstract clearly summarizes the key aspects of the study. To make it more impactful, the authors could highlight the novelty of using EIT-guided PEEP without recruitment manoeuvres in the concluding sentence. They could also add a statement on the clinical implications/importance of the findings. This would help readers understand the impact right away.
Introduction:
- The introduction effectively provides background and rationale for the study. The authors could consider adding a sentence linking the use of individualized PEEP to potential reductions in postoperative complications. This would further enhance the clinical significance.
- The authors could consider expanding on how the study findings could influence perioperative management and outcomes in robotic surgery patients. This would further establish significance.
Methods:
- The authors may consider providing additional details on the robotic surgery procedures performed, as this context is important for generalizability of results.
- The authors could consider adding any sample size/power calculations for secondary outcomes. This would assure readers that the study was adequately powered.
Discussion:
- The discussion thoughtfully examines the study findings and limitations. To augment the clinical implications, the authors could elaborate on how the data support the feasibility of implementing EIT-guided PEEP titration in real-world robotic surgery settings.
- The authors may consider discussing the cost-effectiveness of EIT-guided PEEP, which is relevant for implementation.
In conclusion, I would like to reiterate my appreciation to both the editor and the authors for the opportunity to review this compelling and informative manuscript. I trust that my suggestions will help enhance the clarity, credibility, and relevance of this important work. I would like to wish the authors success in their ongoing research endeavours.
Author Response
Revision Notes
Dear editors and reviewers:
We would like to thank for your kind letter and for reviewers’ constructive comments on our manuscript entitled " Individualized PEEP without recruitment maneuvers improves intraoperative oxygenation: A randomized controlled study". These comments are very valuable and helpful for us to improve our manuscript. All authors have seriously discussed according to all these comments, thereby we have tried our best to revise our manuscript to meet the requirements of your journal. In this revised version, changes in our manuscript within the document were all highlighted in blue.
Please find our point-by-point responses to the comments as follows:
Referee: 2
Comment: The abstract clearly summarizes the key aspects of the study. To make it more impactful, the authors could highlight the novelty of using EIT-guided PEEP without recruitment manoeuvres in the concluding sentence. They could also add a statement on the clinical implications/importance of the findings. This would help readers understand the impact right away.
Response: Thanks for your nice suggestion. The clinical implication of our study was that EIT-guided PEEP without recruitment maneuver could also improve perioperative oxygenation in patients undergoing robot-assisted laparoscopic radical prostatectomy with high risk of perioperative atelectasis formation. We added a statement on the clinical implications that these results may benefit patients with the risk of intraoperative hemodynamic instability caused by recruitment maneuvers.
Line 31 “Significance: Individualized PEEP guided by EIT without intraoperative recruitment maneuvers also improved perioperative oxygenation in patients undergoing robot-assisted laparoscopic radical prostatectomy, which may benefit patients with the risk of intraoperative hemodynamic instability caused by recruitment maneuvers.”
Comment: The introduction effectively provides background and rationale for the study. The authors could consider adding a sentence linking the use of individualized PEEP to potential reductions in postoperative complications. This would further enhance the clinical significance.
Response: Thanks for your nice suggestion. We have added this to the introduction of our manuscript according to your suggestion.
Line 48 “Individual PEEP setting during general anesthesia has been reported to reduce intrapulmonary shunt and postoperative pulmonary complications for elderly patients undergoing spinal surgery in prone position.”
Qian M, Yang F, Zhao L, Shen J and Xie Y 2021 Individualized positive end-expiratory pressure titration on respiration and circulation in elderly patients undergoing spinal surgery in prone position under general anesthesia AM J TRANSL RES 13 13835-44
Comment: The authors could consider expanding on how the study findings could influence perioperative management and outcomes in robotic surgery patients. This would further establish significance.
Response: Thanks for your suggestion. In robotic surgery patients, because of the Trendelenburg position and high-abdominal pressure, individual PEEP combined with recruitment maneuvers was commonly used to improve oxygenation. However, hemodynamic instability was often accompanied by recruitment maneuvers. Our study findings may benefit patients with the risk of intraoperative hemodynamic instability. It also indicated that recruitment maneuvers may be unnecessary for improving oxygenation. We have added this in the discussion of our manuscript as your suggestion.
Lines 278-281: “Our study findings may benefit patients with the risk of intraoperative hemodynamic instability. It also indicated that recruitment maneuvers may be unnecessary for improving oxygenation.”
Comment: The authors may consider providing additional details on the robotic surgery procedures performed, as this context is important for generalizability of results.
Response: Thanks for your suggestion. We have provided additional details on the robotic surgery procedures performed in the methods of our manuscript.
Line 132-134 “All patients were placed in steep 30 degrees Trendelenburg prior to incision. Abdominal access, insufflation and docking of the robot occurred at a pneumoperitoneum pressure of 14 mmHg. All cases were completed using the DavinciTM Xi Robot and AirSealTM iFS device (Intuitive Surgical, Inc, USA).”
Comment: The authors could consider adding any sample size/power calculations for secondary outcomes. This would assure readers that the study was adequately powered.
Response: Thanks for your nice suggestion. We have calculated the power test of secondary outcomes as driving pressure, plateau pressure and Cdyn.
Line 190 “The power test of driving pressure was 0.94, plateau pressure 1, Cdyn 0.99.”
Comment: The discussion thoughtfully examines the study findings and limitations. To augment the clinical implications, the authors could elaborate on how the data support the feasibility of implementing EIT-guided PEEP titration in real-world robotic surgery settings.
Response: Thanks for your valuable comments. We have elaborated the feasibility of implementing EIT-guided PEEP titration in real-world robotic surgery settings in the discussion of our manuscript.
Line 283-294 “Our data showed that high incidence of postoperative hypoxemia (26.9% in control group) was found after robotic-assisted prostatectomy, which extended the PACU stay and increased the costs. With the EIT-guided PEEP titration, PaO2/FiO2 was 14.8 kPa higher than the control group before extubation (71.6 ± 10.7 vs. 56.8 ± 14.1, P = 0.003, Table 2). The improved oxygenation extended into the PACU evidenced by lower incidence of postoperative hypoxemia (3.8% vs. 26.9%, P=0.021). Additionally, EIT-guided PEEP titration was associated with lower driving pressures and better compliance. These improvements can be explained by the distribution of local lung ventilation. Those results indicated that postoperative pulmonary complications may be improved in such patients and prognosis may be also improved. Though we did not follow up the postoperative pulmonary complications, the feasibility and advantages of using EIT in individualized PEEP titration undergoing robot-assisted prostatectomy were demonstrated.”
Line 251-256 “The individualized PEEP titration process may be influenced by surgical maneuvers, in order to avoid this influence, in present study the surgeons were asked to suspend their surgical procedure during PEEP titration. The duration of PEEP titration was about 30 min. Since the surgical robot needed some time to prepare after Trendelenburg position was placed, actual time that costed surgeons waiting to perform operation was limited (at most 5-10 min).”
Comment: The authors may consider discussing the cost-effectiveness of EIT-guided PEEP, which is relevant for implementation.
Response: Thanks for your nice suggestion. We have added this to our manuscript.
“The application of EIT technology requires specialized training and the equipment is expensive, which may require consideration of cost-effectiveness. However, our results showed that the EIT technology improved the intraoperative oxygen and decreased the incidence of postoperative hypoxemia, which may short the patients’ hospital staying and decrease the postoperative comorbidity. The EIT examination itself generates almost no cost (i.e. no consumables and low maintenance cost). The overall cost-effectiveness could be in favor of applying EIT-guided PEEP, especially for high-risk patients prone to lung atelectasis.”
Comment: In conclusion, I would like to reiterate my appreciation to both the editor and the authors for the opportunity to review this compelling and informative manuscript. I trust that my suggestions will help enhance the clarity, credibility, and relevance of this important work. I would like to wish the authors success in their ongoing research endeavours.
Response: Thanks for your affirmation. We deeply appreciate the reviewer’s comments.
Best regards!